# Protective Properties of S-layer Protein 2 from *Lactobacillus crispatus* 2029 against *Candida albicans* Infections

**DOI:** 10.3390/biom13121740

**Published:** 2023-12-04

**Authors:** Vyacheslav M. Abramov, Igor V. Kosarev, Andrey V. Machulin, Tatiana V. Priputnevich, Evgenia I. Deryusheva, Alexander N. Panin, Irina O. Chikileva, Tatiana N. Abashina, Vyacheslav G. Melnikov, Nataliya E. Suzina, Ilia N. Nikonov, Anna A. Akhmetzyanova, Valentin S. Khlebnikov, Vadim K. Sakulin, Raisa N. Vasilenko, Vladimir A. Samoilenko, Alexey B. Gordeev, Gennady T. Sukhikh, Vladimir N. Uversky, Andrey V. Karlyshev

**Affiliations:** 1Federal Service for Veterinary and Phytosanitary Surveillance (Rosselkhoznadzor) Federal State Budgetary Institution “The Russian State Center for Animal Feed and Drug Standardization and Quality” (FGBU VGNKI), 123022 Moscow, Russiavgnki@fsvps.gov.ru (A.N.P.);; 2Kulakov National Medical Research Center for Obstetrics, Gynecology and Perinatology, Ministry of Health, 117997 Moscow, Russia; 3Skryabin Institute of Biochemistry and Physiology of Microorganisms, Federal Research Center “Pushchino Scientific Center for Biological Research of Russian Academy of Science”, Russian Academy of Science, 142290 Pushchino, Russia; 4Institute for Biological Instrumentation, Federal Research Center “Pushchino Scientific Center for Biological Research of Russian Academy of Science”, Russian Academy of Science, 142290 Pushchino, Russia; evgenia.deryusheva@gmail.com; 5Laboratory of Cell Immunity, Blokhin National Research Center of Oncology, Ministry of Health RF, 115478 Moscow, Russia; irinatchikileva@mail.ru; 6Gabrichevsky Research Institute for Epidemiology and Microbiology, 125212 Moscow, Russia; 7Federal State Educational Institution of Higher Professional Education Moscow State Academy of Veterinary Medicine and Biotechnology Named after K.I. Skryabin, 109472 Moscow, Russia; 8Institute of Immunological Engineering, 142380 Lyubuchany, Russiarvasilenko17@inbox.ru (R.N.V.); 9Department of Molecular Medicine, Morsani College of Medicine, University of South Florida, Tampa, FL 33612, USA; vuversky@usf.edu; 10Department of Biomolecular Sciences, School of Life Sciences, Chemistry and Pharmacy, Faculty of Health, Science, Social Care and Education, Kingston University London, Kingston upon Thames KT1 2EE, UK; a.karlyshev@kingston.ac.uk

**Keywords:** S-layer protein 2, *Lactobacillus crispatus*, *Candida albicans*, probiotic bacteria, anti-pathogenic potential, anti-adhesive effect, antifungal agent

## Abstract

Previously, the protective role of the S-layer protein 2 (Slp2) of the vaginal *Lactobacillus crispatus* 2029 (LC2029) strain against foodborne pathogens *Campylobacter jejuni*, *Salmonella enterica* serovar Enteritidis, and *Escherichia coli* O157:H was demonstrated. We demonstrate the new roles of the Slp2-positive LC2029 strain and soluble Slp2 against *C. albicans* infections. We show that LC2029 bacteria can adhere to the surface of the cervical epithelial HeLa cells, prevent their contact with *C. albicans*, and block yeast transition to a pathogenic hyphal form. Surface-bound Slp2 provides the ability for LC2029 to co-aggregate with various *C. albicans* strains, including clinical isolates. *C. albicans*-induced necrotizing epithelial damage is reduced by colonization with the Slp2-positive LC2029 strain. Slp2 inhibits the adhesion of various strains of *C. albicans* to different human epithelial cells, blocks yeast transition to a pathogenic hyphal form, and prevents the colonization and pathogenic infiltration of mucosal barriers. Only Slp2 and LC2029 bacteria stimulate the production of protective human β-defensin 3 in various epithelial cells. These findings support the anti-*Candida albicans* potential of the probiotic LC2029 strain and Slp2 and form the basis for further research on their ability to prevent and manage invasive *Candida* infections.

## 1. Introduction

*Candida albicans* is a member of the normal human microbiome. In most individuals, *C. albicans* is a lifelong, harmless commensal. Under certain circumstances, *C. albicans* turns into a fungal pathobiont capable of causing epithelial cell damage. *C. albicans* is a major fungal pathogen in humans and is the most pathogenic species of the *Candida* genus [1]. *C. albicans* exists in the form of yeast (blastopore) or hyphae, i.e., single, oval-shaped cells or elongated, tubular, and branching filaments that form the mycelium [2,3]. Its pathogenicity is directly correlated with adhesive properties [4], large metabolic adaptability to adverse factors [5], and the ability by invasive (hyphal) form to secrete a cytolytic peptide toxin—candidalysin [6]. This toxin is critical for fungal infections and is a key driver of host cell activation, neutrophil recruitment, and Type 17 immunity [6].

In humans, *C. albicans* can cause vaginal, oral, or intestinal candidiasis and life-threatening systemic infections [7]. Vulvovaginal candidiasis (VVC) is the most widespread. VVC, caused by the *Candida* species, affects an estimated 70–75% of women at least once during their lives. In most countries, *C. albicans* is the major cause of VVC. *C. albicans* was reported to account for 85–95% of yeast strains isolated from the vagina [8]. In addition, 5–8% of women have recurrent vulvovaginal candidiasis (RVVC), which is defined by having four or more episodes every year [9]. VVC and RVVC remain a serious problem in reproductive-age women [10]. The risk factors for VVC include hormonal changes, an immunocompromised state, pregnancy, and antibiotic exposure [8].

Oral candidiasis, known as thrush or oropharyngeal candidiasis (OPC), is a common opportunistic infection of the oral cavity, mainly caused by an overgrowth of *C. albicans* [11,12,13,14]. OPC is common among the elderly (especially those who wear dentures), infants, and people with weakened immune systems or those receiving long-term antibiotic treatment. It may also be associated with systemic diseases, such as diabetes mellitus. In individuals with a defective immune response, the pathogen can spread to the pharynx and the esophagus [15].

Intestinal candidiasis (IC) is a common opportunistic infection of the gastrointestinal tract (GIT) [15,16]. The GIT is the largest source of candidemia in violation of the systemic or local immune functions of the mucous membrane [17,18,19]. The GIT is an important niche in the lifecycle of *C. albicans* because this organism does not have a significant environmental reservoir. *C. albicans* is almost always found associated with humans or other mammals, typically in the GIT, including the oral cavity, genitourinary tract, or skin [20]. In the GIT, *C. albicans* encounters and responds to varying features of the physical environment such as pH, oxygen levels, and nutrient levels [21]. *C. albicans* also responds to the secretions produced in the GIT, such as bile [22]. Thus, *C. albicans* is well adapted for growth in the GIT. Oral treatment with broad-spectrum antibiotics destroys the local intestinal microbiota, which provokes the reproduction of intestinal *C. albicans* and negatively affects the immunological system of patients, thereby contributing to the occurrence of viral infection [23]. Violation of the normal intestinal microbiota leads to the development of low-level inflammation, which contributes to increased fungal adhesion and colonization, with the latter leading to the subsequent development of inflammation and stimulation of the production of proinflammatory cytokines IL-23 and IL-17 in the GIT [16,24]. A vicious circle occurs in the intestine: low-level inflammation stimulates increased adhesion of *C. albicans* on enterocytes, whereas the increased adhesion and colonization of *C. albicans* stimulate inflammation [24]. According to a number of studies, inflammation destroys the intestinal barrier, which creates conditions for the penetration of *C. albicans* into the bloodstream [25,26,27]. This creates conditions for the development of systemic candidiasis.

Systemic or disseminated severe candidiasis develops mainly in patients with HIV/AIDS, as well as in patients with diabetes mellitus, and is characterized by high morbidity and mortality rates [28]. Deaths per annum from systemic fungal infections are greater than the global mortality due to malaria or breast cancer and are similar to deaths due to tuberculosis or HIV [29].

Four main classes of antifungal drugs are used for the treatment of candidiasis: azoles, polyenes, allylamines, and echinocandins. Antifungal agents, such as nystatin, amphotericin B, clotrimazole, and fluconazole, are first-line therapies and remain the main strategy for the prevention and treatment of infections, such as vaginal, oral, or intestinal candidiasis, and life-threatening systemic infections [30,31]. However, oral azoles exhibit numerous side effects [32] caused by drug toxicity and the emergence of drug-resistant strains [33]. Treatment with these drugs can lead to growth in Candida spp. resistance and to the development of RVVC. This disease is a common cause of significant morbidity in women in all strata of society, affecting millions of women worldwide [10,30]. Investigations presented in [34] indicated that fluconazole caused dysbiosis of both mycobiota and bacteriobiota. Among the mycobiota, *Candida* spp. decreased but other fungi, such as Aspergillus, Wallemia, and Epicoccum, increased. Among the bacteriobiota, Bacteroides, Allobaculum, Clostridium, Desulfovibrio, and *Lactobacillus* spp. decreased, while Anaerostipes, Coprococcus, and Streptococcus increased. During recent decades, the high rates of morbidity and mortality caused by fungal infections are associated with the current limited antifungal arsenal and the high toxicity of the drugs [35,36]. These facts emphasize the relevance of developing innovative therapeutic strategies. In this regard, probiotics may represent a promising alternative approach. Several studies have reported the anti-pathogenic potential of probiotic bacteria for the prevention and/or the treatment of OPC [32,37,38,39,40,41] and IC [42,43,44,45,46,47,48,49].

Healthy vaginal microbiota is typically dominated by the *Lactobacillus* species, such as *L. crispatus*, *L. jensenii*, and *L. gasseri* [50,51]. *L. crispatus* is prevailing over the other hydrogen peroxide-producing *Lactobacillus* species [52].

A type I microbiota with the dominance of the *L. crispatus* species is important for maintaining a healthy birth tract [50,51,52,53]. *L. crispatus* species is a microbial biomarker of a healthy vaginal microbiota [54,55]. *L. crispatus* is involved in maintaining the homeostasis of the vaginal environment, where it supports the immune barriers of the vagina without causing inflammation, while at the same time reducing pro-inflammatory cytokines [56,57] that usually increase during bacterial vaginosis, BV [58,59], and vulvovaginal candidiasis, VVC [10]. Strains of *L. crispatus* produce various bacteriocins. In eight genomes of vaginal *L. crispatus* strains isolated from women, the genetic determinants of bacteriocins were characterized by studies in silico [60,61]. Genetic determinants of bacteriocins were characterized in the genomes of seven strains of *L. crispatus* isolated from chicken feces [60]. *L. crispatus* 2029 produces the class III bacteriocin Helveticin-M [62] and secretes high levels of hydrogen peroxide—120 mg/L [56]. Comparative analysis of the genome of *L. crispatus* isolated from various ecological niches shows the ecological adaptation of this species to the environment of the human vagina and may still undergo adaptation to enhance its competitiveness for niche colonization [63,64]. 

Despite the fact that *L. crispatus* species shows ecological adaptation to the human reproductive system, it also effectively implements probiotic and postbiotic properties in the digestive system of humans and animals. Human strain *L. crispatus* 2029 secretes S-layer protein that enhances growth, differentiation, VEGF production, and barrier functions in intestinal epithelial cell line Caco-2 [65]. *L. crispatus* JCM5810, isolated from chicken feces, reduces *Campylobacter jejuni* colonization in chicken intestines [66]. *L. crispatus* ZJ001, isolated from pig intestines produced S-layer protein that inhibited the adhesion of *S. typhimurium* and *E. coli* O157:H7 to HeLa cells used in the experiment as an in vitro cellular biomodel [67].

It is well known that a crucial prerequisite for the *C. albicans* pathogenicity is the initial adhesion to host cells [68] leading to genitourinary, oropharyngeal, or intestinal candidiasis and life-threatening systemic infections [4,69,70]. Studies on the anti-adhesive effects of the culture supernatants of *Lactobacillus* species against *Candida* cells have been previously reported [71,72], and the supernatants capable of inhibiting *C. albicans* hyphae formation by modulating *Lactobacillus* gene expression were studied [71,72]. The culture supernatants of *L. crispatus* JCM 1185 inhibited *C. albicans* adhesion to HeLa cells and *C. albicans* biofilm formation by downregulating biofilm formation-related genes [73]. We have previously demonstrated that soluble S-layer protein 2 (Slp2) reduces the adhesion of foodborne pathogens *Campylobacter jejuni*, *Salmonella enterica* serovar Enteritidis, and *Escherichia coli* O157:H to the intestinal Caco-2 cells [65]. The anti-adhesive properties of Slp2 secreted into the culture medium by the LC2029 strain against *C. albicans* remained mostly unexplored. The goal of the present study was to elucidate the anti-adhesive effects of Slp2 and its role in the protective properties of the LC2029 strain against *C. albicans* infections.

## 2. Materials and Methods

### 2.1. Bacterial Strains and Growth Conditions

The *L. crispatus* 2029 (LC2029) Slp2-positive strain was originally isolated from a vaginal smear of a healthy woman of reproductive age [56]. Genomic sequencing of this strain revealed that the *slp2* gene was responsible for Slp2 synthesis [62]. The strain was deposited at the All-Russian Collection of Microorganisms at the G. K. Skryabin Institute of Biochemistry and Physiology of Microorganisms under registration number VKM B-2727D. The *L. crispatus* (LC1385) Slp2-negative strain was originally isolated from a vaginal smear of a woman of reproductive age with clinical diagnosis: RVVC, bacterial vaginosis (BV) in anamnesis. LC2029 and LC1385 Slp2-negative strains were grown in Man-Rogosa–Sharpe (MRS) broth or an agar containing MRS plates (Himedia, India) at 37 °C in 5% CO_2_ or anaerobically for 48 h. *C. albicans* strains were cultured to the third generation on a Sabouraud dextrose (SD) agar (HiMedia, India) at 37 °C for 72 h and harvested. After resuspension in phosphate buffered saline (PBS), the cells were counted and adjusted to a final concentration of 1 × 10^5^ CFU/mL. A complete list of the microorganisms used in this study and their growth conditions is provided in Table 1.

### 2.2. In Vitro Models of the Human Epithelial Cells

Cervical HeLa epithelial cells were cultured on Dulbecco’s Modified Eagle Medium (DMEM) with 10% heat-inactivated fetal bovine serum (FBC), 1% nonessential amino acid, and 1% glutamine at 37 °C in a 5% CO_2_ in humidified atmosphere. To obtain a monolayer of immature HeLa cells (formed 70–80%), the cells were cultured for 24 h at 37 °C in a 5% CO_2_ humidified atmosphere. To obtain a monolayer of immature HeLa cells (formed 100%), the cells were cultured for 72 h at 37 °C in a 5% CO_2_ humidified atmosphere.

Vaginal Vk2/E6E7 epithelial cells (ATCC #CRL-2616, VA) were cultured in a keratinocyte serum-free medium (KSFM) (Gibco, Waltham, MA, USA) supplemented with 50 mg/mL of bovine pituitary extract (BPE) and 0.1 ng/mL of epidermal growth factor (EGF) at 37 °C with 5% CO_2_. Experiments were carried out in KSFM without supplements.

Buccal TR146 epithelial carcinoma cell line [74], obtained from the European Collection of Authenticated Cell Cultures (ECACC), were cultured in DMEM (Sigma-Aldrich, St. Louis, MO, USA) and supplemented with 10% FBC and 1% penicillin–streptomycin. Serum-free DMEM was used to replace the normal growth medium 24 h before and during the experimentation process.

Intestinal Caco-2 epithelial cells and colon HT-29 epithelial cells were maintained in a culture medium, DMEM, with 10% FBS, 1% nonessential amino acid, and 1% glutamine. Cells were cultured at 37 °C in a 5% CO_2_ humidified atmosphere. 

Human umbilical vein endothelial cells (HUVECs) were obtained from six human donors. Umbilical cords (15 cm) were cut soon after birth and stored in PBS immediately. The endothelial cells were obtained from filling the lumens of umbilical veins with collagen enzyme solution. 

The cells were cultured at 37 °C in a 5% CO_2_ atmosphere in a standard endothelial cell basal medium (EBM) (PanEco, Moscow, Russia) plus endothelial cell-growth medium (EGM) supplements (CAMBREX Bios Science, Inc., Walkersville, MD, USA) and 10% heat-inactivated FBC (Invitrogen, Waltham, MA, USA). After reaching a confluence of 80%, the cells were detached using 0.25% trypsin-EDTA (PanEco, Moscow, Russia). The cells were used by withing three to four passages.

Human lung carcinoma A-549 cells were cultured in Minimum Essential Medium with 10% FBS, 2 mmol/L L-glutamine, 1 mmol/L pyruvate, 10 mmol/L HEPES, and 50 μmol/L penicillin/streptomycin and maintained at 37 °C in a sterile humidified atmosphere of 5% CO_2_.

All epithelial cells used tested negative for mycoplasma.

### 2.3. Slp2 Production

Slp2 was extracted from the LC2029 strain by 5 M LiCl and purified by cation exchange chromatography. The purified Slp2 was examined by SDS-PAGE using 12% polyacrylamide gel and Western blotting analysis. The molecular weight of the purified soluble Slp2 was ~46 kDa [65]. Slp2 concentration in the solution was determined using a Bio-Rad Protein Assay (Bio-Rad Laboratories, Hercules, CA, USA).

### 2.4. Spectrophotometric Co-Aggregation Assay for the Determination of Interactions between the LC2029 Strain and C. albicans Unicellular Yeast Form

The LC2029 strain and Slp2-negative LC1385 strain were grown in MRS broth at 37 °C anaerobically for 48 h. *C. albicans* strains were grown at 37 °C in SD broth at 37 °C in 5% CO_2_ for 48 h. The cultures were harvested by centrifugation at 10,000× *g* for 10 min, washed twice with sterile PBS pH 7.0, and resuspended in PBS. Absorbance (A_600 nm_) of bacterial suspensions was adjusted to 0.60 ± 0.02 in order to standardize the concentration of bacteria (10^7^ to 10^8^ CFU/mL). LC2029 strain suspension or Slp2-negative LC1385 strain suspension (2 mL each) was mixed with 2 mL of *C. albicans* strain suspensions for at least 10 s by a vortex mixer. After 4 h incubation at 37 °C, the suspensions were measured using a spectrophotometer at OD600 (T1800; Hitachi, Tokyo, Japan).

Co-aggregation between the LC2029 strain or Slp2-negative LC1385 strain and the yeast form of *C. albicans* strains was determined by measuring optical density (OD) at 600 nm according to [75]. The percentage of co-aggregation was calculated by following equation:$$Co\text{-}aggregation(\%)=\frac{\left(({OD}_{target}+{OD}_{LC2029})-2\cdot {OD}_{mix}\right)}{\left({OD}_{target}+{OD}_{LC2029}\right)}$$

OD_target_, OD_LC2029_, and OD_mix_ represent the OD measure at 600 nm of individual *C. albicans* strains, the LC2029 strain, and their mixture after incubation for 2 h.

### 2.5. Slp2 Effect on C. albicans Adhesion to Epithelial Cells of Human Biotopes

Monolayers (formed 70–80%) of immature cervical HeLa epithelial cells, vaginal Vk2/E6E7 epithelial cells, buccal TR146 epithelial cells, intestinal Caco-2 epithelial cells, and colon HT-29 epithelial, HUVECs, and lung A-549 cells were obtained by cultivation in an appropriate nutrient medium for 24–48 h at 37 °C in a 5% CO_2_ in humidified atmosphere.

Suspensions of *C. albicans* strains with 5 × 10^7^ CFU/mL were incubated in the presence of Slp2 at concentrations of 10, 50, and 100 µg/mL in PBS for 30 min. The strains of *C. albicans* treated with Slp2 and the control group of *C. albicans* treated with PBS were then applied to the monolayers of cervical HeLa epithelial cells, vaginal Vk2/E6E7 epithelial cells, buccal TR146 epithelial cells, intestinal Caco-2 epithelial cells, and colon HT-29 epithelial, HUVECs, and lung A-549 epithelial cells. The plates were incubated for 1 h at 37 °C under 5% CO_2_. Cell monolayers were washed three times with sterile PBS to remove unbound *C. albicans* strains, fixed with methanol, stained with azure-eosin (Pan Eco, Russia), and examined under a Leica DM 4500B microscope (Leica, Richmond Hill, ON, Canada). The adhesion activity of *C. albicans* strains was quantified using the Leica IM modular applications system (Leica, Canada). The adhesion activity of *C. albicans* strains to epithelial cells was expressed as percentage of epithelial cells containing candida cells on their surface.

### 2.6. Quantification of C. albicans-Mediated Damage of Human Epithelial Cells

The influence of lactobacilli on *C. albicans*-mediated host cell damage was investigated by measuring the release of cytoplasmic human lactate dehydrogenase (LDH), as an indicator for the loss of membrane integrity and a hallmark of necrosis, according to [76,77]. LDH was quantified in the supernatant of infected epithelial call monolayers 24 h post-infection using the Cytotoxicity Detection Kit (Roche, Basel, Switzerland) according to the manufacturer’s instructions. LDH from rabbit muscle (5 mg/mL, Roche) was used to generate a standard curve for the determination of LDH concentrations. The background control level of uninfected epithelial cells was subtracted from the experimental LDH release and usually compared to 100% *C. albicans* single infection.

### 2.7. ELISA

To determine the concentration of HBD-3, the supernatants obtained from epithelial cultures stimulated with the Slp2, LC2029 strain, Slp2-negative ΔLC2029 strain (Slp2 was eliminated with 5 M lithium chloride), and Slp2-negative LC1385 strain for 12 h were collected, centrifuged to separate bacteria, and stored at 80 °C until assayed using an ELISA kit (Phoenix Pharmaceuticals, Inc., Burlingame, CA, USA) following the protocol provided by the manufacturer.

### 2.8. Statistical Analysis

The results were analyzed using a one-way ANOVA and represented means ± standard errors of the means (SEM). Statistically significant differences were accepted at *p* < 0.05.

## 3. Results and Discussion

### 3.1. Inhibitory Effects of the LC2029 Strain on the Transition of the C. albicans Yeast Form to Hyphae

Figure 1A shows the non-pathogenic yeast form of the *C. albicans* ATCC 10231 strain. The non-pathogenic yeast form of *C. albicans* is found in the normal vaginal microbiota of women [8]. Figure 1B shows a monolayer (formed by 100%) of intact immature HeLa cells used as an in vitro biomodel of vaginal dysbiosis when the content of lactobacilli in the vaginal biotope decreases until complete disappearance [10,50]. *C. albicans* cells in a yeast form introduced into the wells with an intact monolayer of HeLa cells quickly adhere to the surface of the intact monolayer and after 2 h of co-cultivation turn them into a pathogenic hyphal form (Figure 1C). The hyphal form of *C. albicans* is not removed from the surface of the HeLa cell monolayers after washing up to three times with PBS (Figure 1C).

Cells in the LC2029 strain added to wells with a monolayer of Hella cells (formed by 100%) effectively adhere to the surface of the monolayer (Figure 1D). A monolayer (formed by 100%) of intact immature HeLa cells with cells of the LC2029 strain adhered to the monolayer surface and was used by us as an in vitro biomodel of normal vaginal microbiota [50]. Figure 1E shows a monolayer of Hela cells formed by 70–80%. In the field of view, LC2029 cells adhered to the surface of HeLa cells in suspension. *C. albicans* cells in yeast form, introduced into the nutrient medium after 30 min, are subjected to co-aggregation with LC2029 cells in suspension (ratio of cells of LC2029 strain/*C. albicans* cells in yeast form was 200/1). After 4 h of joint cultivation, the co-aggregates of LC2029 cells with the yeast form of *C. albicans* cells in suspension are easily removed from the well by washing three times with PBS. In the field of view, only single *Candida* cells in yeast form are visible, associated with LC2029 cells protecting the monolayer of HeLa cells (Figure 1F). There are no *C. albicans* cells in the pathogenic hyphal form in the field of vision (Figure 1F). The LC2029 strain effectively inhibits the transition of the non-pathogenic yeast form of *C. albicans* strain 10231 into the pathogenic hyphal form (Figure 1F). The obtained results indicate that Slp2 fixed on the surface of LC2029 cells is responsible for the co-aggregation of LC2029 cells with *C. albicans* cells and for the ability of LC2029 cells to inhibit the transition of the non-pathogenic yeast form of *C. albicans* into the pathogenic hyphal form. A type I microbiota with the dominance of the *L. crispatus* species is important for maintaining a healthy birth tract [50]. The vaginal Slp-positive strain CTV-05 of *L. crispatus* successfully colonizes the cervicovaginal biotope [78]. The vaginal probiotic *L. crispatus* exhibits strong antifungal effects against VVC in a rat model [79].

### 3.2. LF2029 Co-Aggregation with Unicellular Yeast Form of Different C. albicans Strains

Comparative studies of LC2029 strain and Slp2-negative LC1385 strain сo-aggregative abilities with unicellular yeast form of different *C. albicans* strains have been conducted (Table 2). The LC2029 strain is a strong co-aggregator of unicellular yeast form of different *C. albicans* strains. LC2029 actively co-aggregated with different *C. albicans* strains, including clinical isolates (MD IIE Ca-CI-8, MD IIE Ca-CI-126, MD IIE Ca-CI-98, MD IIE Ca-CI-154). The percentage of co-aggregation ranged from 74.7 ± 4.3% to 84.1 ± 6.5%. The co-aggregation activity of the Slp2-negative LC1385 strain with unicellular yeast form of different *C. albicans* strains was very low. The co-aggregation abilities are characteristic attributes of some probiotic strains belonging to the *Lactobacillaceae* family. Co-aggregation is one of the mechanisms exerted by probiotics to create a competitive micro-environment around the pathogen and limit its adhesion on the host epithelial cells. Co-aggregation of *C. albicans* and lactobacilli may be important in the vagina, especially to reduce the adhesion of the fungus to the vaginal mucosa and to restore the vaginal microbiome by the production of favorable metabolites [80] and the modulation of cytokine expression [53]. Due to these properties of probiotic lactobacilli, the host organism can avoid colonization of the oral cavity, intestines, reproductive tract, and other organs by fungal pathogens [4].

### 3.3. Slp2 Effect on C. albicans Adhesion to the Epithelial Cells of the Human Cervicovaginal Biotope

*C. albicans* is an opportunistic pathogen with a significant ability to interact with epithelial cells of different human biotopes in terms of the adherence, invasion, and induction of cell damage [16]. 

Slp2 isolated from vaginal lactobacilli strain LC2029 very efficiently inhibited the adhesion of *C. albicans* yeast form to human cervical and vaginal epithelial cells (Table 3 and Table 4). Slp2 was equally effective in inhibiting the adhesion of *C. albicans* strains from the collection and clinical isolates, including the Ca-CI-8 strain isolated from a patient with recurrent candidiasis.

Necrosis and inflammation of the cervical epithelium induced by *C. albicans* increase the ability of the human papilloma virus (HPV) to form a malignant epithelial tumor [81]. Attachment of *C. albicans* to epithelial cells is a prerequisite for the colonization and pathogenic infiltration of mucosal barriers [82].

Chemical strategies to reduce physical interaction between pathogenic fungi and host cells show promising results. The synthesis of a multivalent glycoconjugate in which an inhibitor of *C. albicans* adhesion is chemically coupled to a linear peptoid scaffold was described.

Fungal adherence to buccal epithelial cells was reduced in vitro following treatment with glycoconjugate formulation, and investigations to elucidate the precise mechanism of action are currently ongoing [83]. These studies confirm the importance of the anti-adhesive properties of soluble Slp2 of LC2029 in relation to *C. albicans* and the expediency of its use for the prevention of VVC.

### 3.4. Slp2 Effect on C. albicans the Adhesion to Epithelial Cells of the Human Oropharyngeal Biotope

Slp2 very efficiently inhibited the adhesion of *C. albicans* yeast form to human buccal TR146 epithelial cells (Table 5). Slp2 was equally effective in inhibiting the adhesion of *C. albicans* strains from the collection and clinical isolates, including the Ca-CI-126 strain isolated from a patient with OPC. *C. albicans* is a frequent component of oral ecology.

In immunocompromised patients, *C. albicans* can cause a multitude of disease manifestations ranging from mild oral disease to disseminated candidiasis. Diagnosis and treatment of disease caused by *C. albicans* are especially important in HIV/AIDS patients who, despite the advent of antiretroviral therapy (ART), continue to suffer significant *Candida*-associated morbidity [84,85].

### 3.5. Slp2 Effect on C. albicans Adhesion to the Epithelial Cells of the Human Intestinal Biotope

Slp2 inhibited the adhesion of *C. albicans* yeast form to human intestinal Caco-2 cells and HT-29 cells (Table 6 and Table 7). Slp2 was equally effective in inhibiting the adhesion of *C. albicans* strains from the collection and clinical isolates, including Ca-CI-98 strain isolated from a patient with IC.

In healthy individuals *C. albicans* is predominantly found as part of the gastrointestinal microbiome. With IC, *C. albicans* turns into a fungal pathobiont that is able to cause epithelial cell necrosis and is immune activation and intestinal inflammation [23,86]. Intestinal inflammation is known to destroy the intestinal barrier [87]. This creates conditions for the entry of *C. albicans* into the blood and the development of systemic candidiasis [2,88]. We have previously shown that the Slp2 and LC2029 strain inhibited proinflammatory cytokine Il-8 production in Caco-2 and HT-29 cells induced by MALP-2 (agonist of TLR2) and increased the production of anti-inflammatory cytokine Il-6 [57]. Slp2 inhibited the production of CXCL1 and RANTES by Caco-2 cells during differentiation and maturation [57]. Slp2 stimulated VEGF production, decreased paracellular permeability, and increased transepithelial electrical resistance, strengthening barrier function of Caco-2 cell monolayers [65]. The results of these studies indicate the prospects of using the Slp2 and LC2029 strain for the prevention of systemic or disseminated severe candidiasis.

### 3.6. Slp2 Effect on C. albicans Adhesion to HUVECs and Lung A-549 Epithelial Cells

During systemic or disseminated severe candidiasis in susceptible individuals, *C. albicans* gains access to the bloodstream and various internal organs, including the lungs [2]. The intestine–lung axis functions in the human body [89]. During systemic candidiasis, *C. albicans* cells can spread to the lungs, which potentially differ in the availability of nutrients compared to the digestive or reproductive systems [2]. As an in vitro biomodel of systemic candidiasis, we used HUVECs and lung epithelial A-549 cells to study the ability of Slp2 to inhibit the adhesion of various *C. albicans* strains to these cells. Slp2 inhibited the adhesion of *C. albicans* yeast form to HUVECs and A-549 cells (Table 8 and Table 9). Slp2 was equally effective in inhibiting the adhesion of *C. albicans* strains from the collection and clinical isolates, including the Ca-CI-154 strain clinical bloodstream isolate from a patient with systemic candidiasis.

The use of molecular typing methods to compare strains isolated from a patient’s blood with strains isolated from other body sites has shown that *C. albicans* blood isolates are often identical to rectal isolates [18]. These findings support the model that commensal organisms residing in the GIT can escape from this niche and reach the bloodstream. The entry of *C. albicans* strains into the blood occurs after the destruction of the intestinal barrier [87].

The ability of Slp2 to inhibit the adhesion of *C. albicans* strains to HUVECs and lung A-549 epithelial cells suggests a possible application of this soluble protein for the complex treatment of systemic candidiasis.

### 3.7. Reducing C. albicans-Induced Necrotizing Epithelial Damage by Colonization with the LC 2029 Strain

The necrotizing potential of producing the candidalysin *C. albicans* ATCC 10231 strain against different human epithelial cells was assessed by an increase in epithelial cytosolic LDH secreted into the culture medium. The results of the studies are shown in Table 10. The Slp2-negative LC1385 strain and LC2029 strain do not produce LDH and are non-toxic for HUVECs and epithelial cells from a cervicovaginal biotope, including HeLa cells and Vk2/E6E7 vaginal epithelial cells from an oropharyngeal biotope, buccal TR146 epithelial cells from an intestinal biotope, Caco-2 cells and HT-29 from a lung biotope, and A-549 cells. Colonization with the Slp2-negative LC1385 strain before *C. albicans* infection had no effect on *Candida*-induced damage (Table 10). Colonization with the LC 2029 strain before *C. albicans* infection reduced the *C. albicans*-induced necrotic cytotoxicity of epithelial cells. It is known that candidalysin is secreted by *C. albicans* cells that have passed into the pathogenic hyphal form and adhered to the surface of epithelial cells [90]. Secreted candidalysin is readily assembled into polymers in a solution and binds to the plasma membrane of the epithelial cell, forms membrane pores, and causes its necrotic lesion [6,91]. The plasma membrane is central for homeostatic maintenance in mammalian cells [77,92]. Necrotic cell death induces inflammatory responses and is closely associated with inflammatory diseases [93]. The results obtained indicate the ability of the LC2029 strain, but not the Slp2-negative LC1385 strain, to inhibit the secretion of candidalysin by the *C. albicans* ATCC 10231 strain and prevent necrotic damage to epithelial cells of various human biotopes.

### 3.8. Slp2 and LC2029 Strain Stimulate HBD-3 Production in Epithelial Cells of Various Biotopes

Slp2 and LC2029 bacteria, but not Slp2-negative ΔLC2029 bacteria (Slp2 was eliminated with 5 M lithium chloride) and Slp2-negative LC1385 bacteria, stimulate the production of protective antimicrobial peptide (AMP) HBD-3 in epithelial cells belonging to various human biotopes (Table 11). It is known that β-defensins induce cationic peptides produced by epithelial cells that have been proposed to be an important component of immune function on mucosal surfaces [94]. It is known that HBD-3 significantly reduces *C. albicans* viability and growth [53] and serves as an innate defense against *Candida* invasion on human mucosal surfaces.

## 4. Conclusions

The adhesion of *C. albicans* to the mucosal surfaces is the basis of its invasion into the body, which occurs already in the first minutes of the pathogen’s interaction with the epithelial cells of the vagina, oral cavity, or intestines. From this moment on, vaginal, oropharyngeal, or intestinal candidiasis develops in the macroorganism. Using cervical epithelial HeLa cells as a model for the in vitro cervical biotope, it has been shown that Slp2 fixed on the surface of LC2029 cells is responsible for the co-aggregation of LC2029 cells with yeast form of *C. albicans* cells and for the ability of LC2029 cells to inhibit the transition of the non-pathogenic yeast form of *C. albicans* into the pathogenic hyphal form. Soluble Slp2 (postbiotics) inhibits *C. albicans* yeast form adhesion to HeLa cells. Using human Vk2/E6E7 vaginal epithelial cells as a model for the in vitro vaginal biotope, it has been shown that soluble Slp2 inhibits various strains of *C. albicans* yeast form adhesion to these cells. Using human buccal TR 146 epithelial cells as a model for the in vitro oropharyngeal biotope, it has been shown that soluble Slp2 inhibits various strains of *C. albicans* yeast form adhesion to these cells. Using human intestinal Caco-2 and HT-29 epithelial cells as a model for the in vitro intestinal biotope, it has been shown that soluble Slp2 inhibits various strains of *C. albicans* yeast form adhesion to these cells. Using human HUVECs and A549 epithelial cells as a model for the in vitro systemic candidiasis, it has been shown that soluble Slp2 inhibits various strains of *C. albicans* yeast form adhesion to these cells. The Slp2-positive LC2029 strain, but not the Slp2-negative LC1385 strain, protects HeLa, Vk2/E6E7, TR146, Caco-2, HT-29, HUVECs, and A549 epithelial cells from *C. albicans*-induced necrotizing epithelial damage. Soluble Slp2 and Slp2-positive LC2029 bacteria stimulate the production of anti-Candida protective HBD-3 in epithelial cells belonging to the various biotopes. Further research will be devoted to the study of the mechanisms associated with the ability of Slp2-positive LC2029 and soluble Slp2 to inhibit the transition of the non-pathogenic yeast form of *C. albicans* into the pathogenic hyphal form.

## Figures and Tables

**Figure 1 biomolecules-13-01740-f001:**
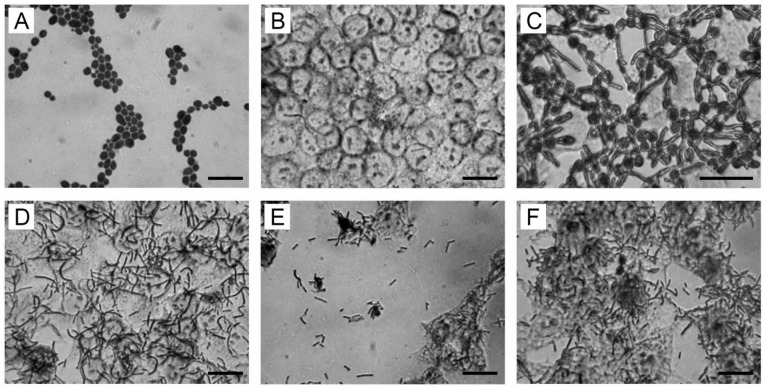
Inhibitory effects of the LC2029 strain on the transition of *C. albicans* yeast form to hyphal. Optical microscopy (bar—10 µm). (**A**) Yeast form of *C. albicans* strain ATCC 10231. (**B**) Monolayer (formed by 100%) of intact immature HeLa cells. (**C**) Intensive transition of *C. albicans* yeast form to hyphal on intact monolayer of HeLa cells. All cells of the *C. albicans* yeast form, after being added to the intact HeLa monolayer, turned into hyphal form after 2 h of joint cultivation. (**D**) The monolayer (formed by100%) of immature HeLa cells is protected by cells of strain LC2029 in the adhered state. (**E**) The monolayer (formed by 70–80%) of immature HeLa cells is protected by cells of the LC2029 strain in the adhered state. Cells of the LC2029 strain are also visible in the nutrient medium in a suspension state. Thirty minutes after the introduction of the *C. albicans* in yeast form into the wells (the ratio of LC2029 strain/*C. albicans* yeast form is 100/1 CFU/mL), the cells of the LC2029 strain in the nutrient medium in suspension form co-aggregates with the *C. albicans* yeast form. (**F**) Co-aggregates of strain LC2029 cells with *C. albicans* cells in yeast form are removed when the culture medium is replaced or washed once by PBS. Only single *C. albicans* cells in yeast form are visible in the field of view HeLa cells.

**Table 1 biomolecules-13-01740-t001:** The microorganisms used in this study.

Microorganism	Strain	Growth Conditions
*L. crispatus*	LC2029 ^1^	MRS ^a^, 37 °C in a CO_2_ incubator, 10% CO_2_ or anaerobically 48 h
*L. crispatus*	LC1385 ^2^	The same
*C. albicans*	ATCC ^3^ 10231	SD ^b^ 37 °C in 5% CO_2_ 24–48 h
*C. albicans*	ATCC 11006	The same
*C. albicans*	ATCC 2091	The same
*C. albicans*	ATCC 64548	The same
*C. albicans*	MD IIE Ca-CI-8 ^4^	The same
*C. albicans*	MD IIE Ca-CI-126 ^5^	The same
*C. albicans*	MD IIE Ca-CI-98 ^6^	The same
*C. albicans*	MD IIE Ca-CI-154 ^7^	The same

^1^ Russian Collection of Microorganisms at the Skryabin Institute of Biochemistry and Physiology of Microorganisms, Federal Research Center “Pushchino Scientific Center for Biological Research, RAS” Pushchino, Moscow Region, Russia. ^2^ Collection of Microorganisms at the Institute of Immunological Engineering, Department of Biochemistry of Immunity and Biodefence, Lyubuchany, Moscow Region, Russia. ^3^ American Type Culture Collection, Manassas, VA, USA. ^4^ Clinical isolates (recurrent VVC). ^5^ Clinical isolates (OPC). ^6^ Clinical isolates (IC). ^7^ Clinical bloodstream isolates. ^a^ MRS broth or an agar containing plates (HiMedia, India). ^b^ SD broth or an agar containing plates (HiMedia, India).

**Table 2 biomolecules-13-01740-t002:** Co-aggregative abilities of the LC2029 strain and Slp2-negative LC1385 strain with unicellular yeast form of *C. albicans* strains.

Strain	LC2029	LC1385
*C. albicans* ATCC 10231	76.1 ± 4.6 ***	3.0 ± 0.1
*C. albicans* ATCC 11006	82.8 ± 5.4 ***	2.1 ± 0.5
*C. albicans* ATCC 2091	79.3 ± 4.8 ***	4.2 ± 0.7
*C. albicans* ATCC 64548	74.7 ± 4.3 ***	2.9 ± 0.5
*C. albicans* MD IIE Ca-CI-8	82.0 ± 5.5 ***	3.0 ± 0.2
*C. albicans* MD IIE Ca-CI-126	78.2 ± 5.2 ***	2.2 ± 0.4
*C. albicans* MD IIE Ca-CI-98	84.1 ± 6.5 ***	3.4 ± 0.7
*C. albicans* MD IIE Ca-CI-154	78.6 ± 4.8 ***	2.1 ± 0.5

*** *p* < 0.001—сo-aggregation of the LC2029 strain with *C. albicans* strains vs. сo-aggregation of the LC1385 strain with *C. albicans* strains. All data are representative of six independent experiments tested in triplicate.

**Table 3 biomolecules-13-01740-t003:** Slp2 reduction in the *C. albicans* strains in the process of adhesion to HeLa cells.

Strain	HeLa Cells Containing Adhered Bacteria, %			
	Concentration of Slp2 in a Medium, µg/mL			
	0	10	50	100
*C. albicans* ATCC 10231	67.0 ± 4.6	35.1 ± 3.8 *	12.0 ± 1.3 **	2.1 ± 0.4 ***
*C. albicans* ATCC 11006	65.4 ± 3.8	39.0 ± 3.6 *	14.2 ± 1.5 **	1.9 ± 0.6 ***
*C. albicans* ATCC 2091	75.5 ± 4.5	42.2 ± 3.7 *	15.1 ± 1.7 **	3.2 ± 0.4 ***
*C. albicans* ATCC 64548	69.1 ± 5.2	30.0 ± 3.5 *	12.9 ± 1.2 **	1.7 ± 0.3 ***
*C. albicans* MD IIE Ca-CI-8	87.0 ± 2.7	38.4 ± 3.2 *	15.5 ± 2.0 **	3.0 ± 0.5 ***
*C. albicans* MD IIE Ca-CI-126	75.2 ± 3.5	36.7 ± 4.5 *	13.1 ± 1.9 **	3.1 ± 0.6 ***
*C. albicans* MD IIE Ca-CI-98	63.4 ± 2.6	36.2 ± 4.2 *	15.0 ± 3.7 **	3.2 ± 0.5 ***
*C. albicans* MD IIE Ca-CI-154	64.7 ± 3.4	29.1 ± 3.7 *	17.4 ± 4.2 **	1.9 ± 0.4 ***

* *p* < 0.05 adhesion of strains (*C. albicans* ATCC 10231, *C. albicans* ATCC 11006, *C. albicans* ATCC 2091, *C. albicans* ATCC 64548, *C. albicans* MD IIE Ca-CI-8, *C. albicans* MD IIE Ca-CI-126, *C. albicans* MD IIE Ca-CI-98, *C. albicans* MD IIE Ca-CI-154) to HeLa cells alone vs. adhesion to HeLa cells + Slp2 (10 µg/mL); ** *p* < 0.01 adhesion of strains to HeLa cells alone vs. adhesion to HeLa cells + Slp2 (50 µg/mL); *** *p* < 0.001 adhesion of strains to HeLa cells alone vs. adhesion to HeLa cells + Slp2 (100 µg/mL). Data represent the mean and SEM of six independent experiments tested in triplicate.

**Table 4 biomolecules-13-01740-t004:** Slp2 reduction in the *C. albicans* strains in the process of adhesion to human Vk2/E6E7 vaginal epithelial cells.

Strain	Vk2/E6E7 Cells Containing Adhered Bacteria, %			
	Concentration of Slp2 in a Medium, µg/mL			
	0	10	50	100
*C. albicans* ATCC 10231	64.1 ± 5.5	29.1 ± 3.6 *	10.2 ± 1.4 **	3.0 ± 0.7 ***
*C. albicans* ATCC 11006	73.4 ± 4.6	42.4 ± 4.1 *	13.2 ± 1.7 **	2.1 ± 0.4 ***
*C. albicans* ATCC 2091	68.5 ± 3.7	28.1 ± 2.5 *	12.3 ± 1.5 **	2.2 ± 0.3 ***
*C. albicans* ATCC 64548	65.0 ± 6.2	26.8 ± 3.6 *	13.8 ± 1.3 **	2.8 ± 0.6 ***
*C. albicans* MD IIE Ca-CI-8	54.2 ± 3.9	25.0 ± 2.8 *	11.1 ± 1.5 **	2.2 ± 0.4 ***
*C. albicans* MD IIE Ca-CI-126	63.2 ± 4.9	36.1 ± 4.5 *	9.0 ± 1.2 **	3.9 ± 0.5 ***
*C. albicans* MD IIE Ca-CI-98	56.1 ± 2.5	33.6 ± 3.7 *	12.4 ± 1.0 **	4.3 ± 0.8 ***
*C. albicans* MD IIE Ca-CI-154	66.8 ± 5.3	38.0 ± 4.6 *	12.6 ± 1.5 **	1.8 ± 0.3 ***

* *p* < 0.05 adhesion of strains (*C. albicans* ATCC 10231, *C. albicans* ATCC 11006, *C. albicans* ATCC 2091, *C. albicans* ATCC 64548, *C. albicans* MD IIE Ca-CI-8, *C. albicans* MD IIE Ca-CI-126, *C. albicans* MD IIE Ca-CI-98, *C. albicans* MD IIE Ca-CI-154) to Vk2/E6E7 cells alone vs. adhesion to Vk2/E6E7 cells + Slp2 (10 µg/mL); ** *p* < 0.01 adhesion of strains to Vk2/E6E7 cells alone vs. adhesion to Vk2/E6E7 cells + Slp2 (50 µg/mL); *** *p* < 0.001 adhesion of strains to Vk2/E6E7 cells alone vs. adhesion to Vk2/E6E7 cells + Slp2 (100 µg/mL). Data represent the mean and SEM of six independent experiments tested in triplicate.

**Table 5 biomolecules-13-01740-t005:** Slp2 reduction in the *C. albicans* strains in the process of adhesion to buccal TR146 epithelial cells.

Strain	TR146 Cells Containing Adhered Bacteria, %			
	Concentration of Slp2 in a Medium, µg/mL			
	0	10	50	100
*C. albicans* ATCC 10231	54.2 ± 5.5	28.1 ± 4.3 *	9.1 ± 1.6 **	2.1 ± 0.5 ***
*C. albicans* ATCC 11006	57.5 ± 5.3	28.8 ± 3.6 *	10.0 ± 1.8 **	3.0 ± 0.4 ***
*C. albicans* ATCC 2091	74.9 ± 4.6	33.8 ± 2.5 *	10.0 ± 1.3 **	3.2 ± 0.2 ***
*C. albicans* ATCC 64548	55.8 ± 5.3	38.0 ± 4.2 *	11.8 ± 1.6 **	3.9 ± 0.5 ***
*C. albicans* MD IIE Ca-CI-8	77.8 ± 5.6	41.2 ± 3.8 *	10.1 ± 1.5 **	3.1 ± 0.3 ***
*C. albicans* MD IIE Ca-CI-126	59.1 ± 4.3	35.2 ± 2.6 *	13.9 ± 2.7 **	2.0 ± 0.4 ***
*C. albicans* MD IIE Ca-CI-98	68.1 ± 4.6	39.0 ± 3.5 *	11.0 ± 1.8 **	2.8 ± 0.5 ***
*C. albicans* MD IIE Ca-CI-154	52.2 ± 4.7	34.3 ± 2.8 *	11.8 ± 1.5 **	4.0 ± 0.6 ***

* *p* < 0.05 adhesion of strains (*C. albicans* ATCC 10231, *C. albicans* ATCC 11006, *C. albicans* ATCC 2091, *C. albicans* ATCC 64548, *C. albicans* MD IIE Ca-CI-8, *C. albicans* MD IIE Ca-CI-126, *C. albicans* MD IIE Ca-CI-98, *C. albicans* MD IIE Ca-CI-154) to TR146 cells alone vs. adhesion to TR146 cells + Slp2 (10 µg/mL); ** *p* < 0.01 adhesion of strains to TR146 cells alone vs. adhesion to TR146 cells + Slp2 (50 µg/mL); *** *p* < 0.001 adhesion of strains to TR146 cells alone vs. adhesion to TR146 cells + Slp2 (100 µg/mL). Data represent the mean and SEM of six independent experiments tested in triplicate.

**Table 6 biomolecules-13-01740-t006:** Slp2 reduction in the *C. albicans* strains in the process of adhesion to intestinal Caco-2 cells.

Strain	Caco-2 Cells Containing Adhered Bacteria, %			
	Concentration of Slp2 in a Medium, µg/mL			
	0	10	50	100
*C. albicans* ATCC 10231	78.4 ± 3.5	29.1 ± 2.6 *	11.1 ± 2.1 **	4.1 ± 0.3 ***
*C. albicans* ATCC 11006	63.0 ± 3.0	31.5 ± 2.5 *	13.2 ± 2.6 **	3.2 ± 0.2 ***
*C. albicans* ATCC 2091	75.1 ± 4.5	40.7 ± 3.4 *	15.0 ± 3.2 **	2.0 ± 0.3 ***
*C. albicans* ATCC 64548	56.2 ± 2.3	40.0 ± 2.6 *	14.0 ± 2.8 **	3.9 ± 0.5 ***
*C. albicans* MD IIE Ca-CI-8	73.1 ± 3.8	31.9 ± 3.7 *	12.4 ± 2.5 **	2.9 ± 0.4 ***
*C. albicans* MD IIE Ca-CI-126	63.8 ± 2.6	41.2 ± 3.8 *	10.9 ± 2.3 **	2.0 ± 0.3 ***
*C. albicans* MD IIE Ca-CI-98	64.7 ± 4.0	27.4 ± 4.3 *	15.9 ± 3.1 **	3.1 ± 0.5 ***
*C. albicans* MD IIE Ca-CI-154	72.7 ± 3.8	30.6 ± 3.6 *	11.8 ± 2.4 **	3.3 ± 0.4 ***

* *p* < 0.05 adhesion of strains (*C. albicans* ATCC 10231, *C. albicans* ATCC 11006, *C. albicans* ATCC 2091, *C. albicans* ATCC 64548, *C. albicans* MD IIE Ca-CI-8, *C. albicans* MD IIE Ca-CI-126, *C. albicans* MD IIE Ca-CI-98, *C. albicans* MD IIE Ca-CI-154) to Caco-2 alone vs. adhesion to Caco-2 + Slp2 (10 µg/mL); ** *p* < 0.01 adhesion of strains to Caco-2 alone vs. adhesion to Caco-2 + Slp2 (50 µg/mL); *** *p* < 0.001 adhesion of strains to Caco-2 alone vs. adhesion to Caco-2 + Slp2 (100 µg/mL). Data represent the mean and SEM of six independent experiments tested in triplicate.

**Table 7 biomolecules-13-01740-t007:** Slp2 reduction in the *C. albicans* strains in the process of adhesion to intestinal HT-29 cells.

Strain	HT-29 Cells Containing Adhered Bacteria, %			
	Concentration of Slp2 in a Medium, µg/mL			
	0	10	50	100
*C. albicans* ATCC 10231	59.2 ± 3.5	36.1 ± 2.8 *	9.0 ± 1.4 **	2.9 ± 0.2 ***
*C. albicans* ATCC 11006	66.5 ± 3.2	28.8 ± 2.5 *	12.1 ± 2.1 **	2.1 ± 0.1 ***
*C. albicans* ATCC 2091	71.0 ± 3.6	32.2 ± 3.2 *	13.8 ± 2.5 **	3.8 ± 0.5 ***
*C. albicans* ATCC 64548	64.8 ± 4.8	36.7 ± 2.7 *	11.2 ± 2.0 **	2.0 ± 0.1 ***
*C. albicans* MD IIE Ca-CI-8	58.0 ± 4.2	41.0 ± 3.1 *	9.5 ± 1.2 **	2.0 ± 0.1 ***
*C. albicans* MD IIE Ca-CI-126	71.1 ± 3.8	35.1 ± 2.6 *	10.8 ± 1.8 **	3.1 ± 0.2 ***
*C. albicans* MD IIE Ca-CI-98	67.8 ± 3.7	42.9 ± 2.8 *	12.6 ± 1.9 **	2.0 ± 0.4 ***
*C. albicans* MD IIE Ca-CI-154	73.0 ± 4.0	36.0 ± 3.5 *	12.0 ± 2.1 **	2.9 ± 0.3 ***

* *p* < 0.05 adhesion of strains (*C. albicans* ATCC 10231, *C. albicans* ATCC 11006, *C. albicans* ATCC 2091, *C. albicans* ATCC 64548, *C. albicans* MD IIE Ca-CI-8, *C. albicans* MD IIE Ca-CI-126, *C. albicans* MD IIE Ca-CI-98, *C. albicans* MD IIE Ca-CI-154) to HT-29 alone vs. adhesion to HT-29 + Slp2 (10 µg/mL); ** *p* < 0.01 adhesion of strains to HT-29 alone vs. adhesion to HT-29+ Slp2 (50 µg/mL); *** *p* < 0.001 adhesion of strains to HT-29 alone vs. adhesion to HT-29 + Slp2 (100 µg/mL). Data represent the mean and SEM of six independent experiments tested in triplicate.

**Table 8 biomolecules-13-01740-t008:** Slp2 reduction in the *C. albicans* strains in the process of adhesion to HUVECs.

Strain	HUVECs Containing Adhered Bacteria, %			
	Concentration of Slp2 in a Medium, µg/mL			
	0	10	50	100
*C. albicans* ATCC 10231	72.3 ± 4.3	35.6 ± 2.1 *	14.0 ± 1.0 **	2.1 ± 0.1 ***
*C. albicans* ATCC 11006	69.1 ± 3.6	34.1 ± 2.5 *	11.2 ± 0.9 **	1.8 ± 0.3 ***
*C. albicans* ATCC 2091	72.7 ± 4.1	40.0 ± 2.3 *	10.0 ± 0.8 **	2.0 ± 0.1 ***
*C. albicans* ATCC 64548	64.5 ± 3.4	32.3 ± 2.4 *	11.7 ± 1.2 **	3.1 ± 0.2 ***
*C. albicans* MD IIE Ca-CI-8	57.7 ± 3.2	35.1 ± 2.1 *	11.0 ± 1.0 **	2.0 ± 0.1 ***
*C. albicans* MD IIE Ca-CI-126	71.0 ± 3.5	38.0 ± 2.2 *	12.9 ± 1.1 **	2.8 ± 0.5 ***
*C. albicans* MD IIE Ca-CI-98	69.1 ± 4.1	29.7 ± 2.7 *	11.1 ± 0.9 **	2.0 ± 0.1 ***
*C. albicans* MD IIE Ca-CI-154	67.0 ± 3.8	39.1 ± 2.5 *	15.0 ± 1.4 **	3.8 ± 0.6 ***

* *p* < 0.05 adhesion of strains (*C. albicans* ATCC 10231, *C. albicans* ATCC 11006, *C. albicans* ATCC 2091, *C. albicans* ATCC 64548, *C. albicans* MD IIE Ca-CI-8, *C. albicans* MD IIE Ca-CI-126, *C. albicans* MD IIE Ca-CI-98, *C. albicans* MD IIE Ca-CI-154) to HUVECs alone vs. adhesion to HUVECs + Slp2 (10 µg/mL); ** *p* < 0.01 adhesion of strains to HUVECs alone vs. adhesion to HUVECs + Slp2 (50 µg/mL); *** *p* < 0.001 adhesion of strains to HUVECs alone vs. adhesion to HUVECs + Slp2 (100 µg/mL). Data represent the mean and SEM of six independent experiments tested in triplicate.

**Table 9 biomolecules-13-01740-t009:** Slp2 reduction in the *C. albicans* strains in the process of adhesion to lung A-549 epithelial cells.

Strain	A-549 Cells Containing Adhered Bacteria, %			
	Concentration of Slp2 in a Medium, µg/mL			
	0	10	50	100
*C. albicans* ATCC 10231	79.6 ± 4.6	37.1 ± 2.9 *	16.0 ± 2.8 **	3.1 ± 0.5 ***
*C. albicans* ATCC 11006	76.1 ± 3.9	35.0 ± 3.1 *	16.8 ± 3.1 **	2.0 ± 0.1 ***
*C. albicans* ATCC 2091	82.1 ± 4.3	38.0 ± 3.5 *	15.2 ± 2.4 **	2.0 ± 0.4 ***
*C. albicans* ATCC 64548	78.9 ± 5.4	42.7 ± 3.6 *	13.5 ± 2.2 **	2.9 ± 0.2 ***
*C. albicans* MD IIE Ca-CI-8	87.2 ± 4.2	39.4 ± 4.8 *	18.8 ± 2.5 **	3.2 ± 0.4 ***
*C. albicans* MD IIE Ca-CI-126	84.3 ± 3.9	42.0 ± 3.5 *	15.0 ± 3.4 **	2.1 ± 0.2 ***
*C. albicans* MD IIE Ca-CI-98	83.2 ± 4.7	35.1 ± 4.2 *	18.1 ± 2.3 **	1.1 ± 0.2 ***
*C. albicans* MD IIE Ca-CI-154	88.7 ± 4.5	41.1 ± 3.7 *	17.4 ± 2.6 **	2.0 ± 0.3 ***

* *p* < 0.05 adhesion of strains (*C. albicans* ATCC 10231, *C. albicans* ATCC 11006, *C. albicans* ATCC 2091, *C. albicans* ATCC 64548, *C. albicans* MD IIE Ca-CI-8, *C. albicans* MD IIE Ca-CI-126, *C. albicans* MD IIE Ca-CI-98, *C. albicans* MD IIE Ca-CI-154) to A-549 alone vs. adhesion to A-549 + Slp2 (10 µg/mL); ** *p* < 0.01 adhesion of strains to A-549 alone vs. adhesion to A-549 + Slp2 (50 µg/mL); *** *p* < 0.001 adhesion of strains to A-549 alone vs. adhesion to A-549 + Slp2 (100 µg/mL). Data represent the mean and SEM of six independent experiments tested in triplicate.

**Table 10 biomolecules-13-01740-t010:** Colonization of epithelial cells with the LC2029 strain reduces *C. albicans*-induced necrosis.

Epithelial Cells	LDH (ng/mL)					
	Medium	LC1385	LC2029	*C. albicans*	*C. albicans* + LC1385	*C. albicans* + LC2029
HeLa	40 ± 4	41 ± 4	41 ± 4	118 ± 5 **	123 ± 5 **	43 ± 4
Vk2/E6EF	50 ± 4	53 ± 4	53 ± 3	123 ± 7 **	114 ± 5 **	53 ± 3
TR 146	35 ± 4	38 ± 4	38 ± 3	138 ± 5 **	128 ± 6 **	40 ± 4
Caco-2	59 ± 5	60 ± 3	60 ± 4	145 ± 8 **	133 ± 8 **	63 ± 4
HT-29	53 ± 4	53 ± 3	53 ± 4	147 ± 5 **	138 ± 6 **	50 ± 5
A-549	38 ± 5	38 ± 5	38 ± 6	135 ± 6 **	126 ± 6 **	43 ± 4
HUVECs	38 ± 3	40 ± 4	40 ± 5	116 ± 7 **	115 ± 5 **	41 ± 3

** *p* < 0.01—LDH production in Hela cells, Vk2/E6E7 cells, TR 146 cells, Caco-2 cells, HT-29 cells, A-549 cells, HUVECs induced by *C. albicans* ATCC 10231 and *C. albicans* + LC1385 vs. the medium, Slp2-negative LC1385 strain, Slp2-negative ΔLC2029 strain. Data represent the mean and SEM of six independent experiments tested in triplicate.

**Table 11 biomolecules-13-01740-t011:** Production of HBD-3 in epithelial cells from various human biotopes.

Epithelial Cells	HBD-3 (pg/mL)				
	Medium	LC1385	ΔLC2029	LC2029	Slp2
HeLa	3 ± 1	3 ± 2	17 ± 3	157 ± 16 ***	174 ±19 ***
Vk2/E6E7	3 ± 2	3 ± 2	13 ± 3	165 ± 14 ***	168 ± 17 ***
TR 146	7 ± 3	7 ± 3	17 ± 5	153 ±15 ***	164 ± 15 ***
Caco-2	7 ± 3	7 ± 3	13 ± 5	169 ± 18 ***	175 ±16 ***
HT-29	5 ± 3	5 ± 3	17 ± 6	172 ±16 ***	183 ± 14 ***
A-549	3 ± 2	3 ± 2	10 ± 5	97 ± 8 **	94 ± 7 **
HUVECs	3 ± 1	3 ± 1	13 ± 3	83 ± 5 **	80 ± 6 **

** *p* < 0.01—production of HBD-3 in A-549 cells and HUVECs induced by the Slp2 and LC2029 strain vs. the medium, Slp2-negative LC1385 strain, Slp2-negative ΔLC2029 strain; *** *p* < 0.001—production of HBD-3 in HeLa cells, Vk2/E6E7 cells, TR 146 cells, Caco-2 cells, HT-29 cells induced by the Slp2 and LC2029 strain vs. the medium, Slp2-negative LC1385 strain, Slp2-negative ΔLC2029 strain. Data represent the mean and SEM of six independent experiments tested in triplicate.

## Data Availability

Data are contained within the article.

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
