# Peer review of "Protective Properties of S-layer Protein 2 from Lactobacillus crispatus 2029 against Candida albicans Infections"

_biomolecules, 2023, doi:10.3390/biom13121740_

Round 1
Reviewer 1 Report
Comments and Suggestions for Authors
In this manuscript, authors demonstrated protective properties against Candida albicans of S-layer proteins 2 (Slp2) form Lactobacillus crispatus 2029 (LC2029), as previously Slp2 of LC2029 was found to show protective effects against foodborne pathogens. This study was conducted well, and the results supported the protective effects of LC2029, which could be attributed to Slp2. The reviewer has some minor points for further revision as below.
In Table 1, the strain number (2029) and the abbreviation (LC2029) of the target strain are not found but should be included. In addition, the abbreviation of the Slp2-2 negative strain (LC1385) should be shown in Table 1.
So many tables are presented as results. Some of them can be made together in new tables or figures? It seems better to grasp and understand the results.
How was Slp2 prepared from LC2029 and quantified? It should be described. These might be described in previous papers but should be mentioned by referring them even briefly.
Tables 2 and 9; ‘C. albicans strains’ should be ‘Strain’ as other tables.
Reviewer 2 Report
Comments and Suggestions for Authors
Using cervical epithelial HeLa cells as a model for the in vitro experiments, it was revealed that the adhered on the surface of these epithelial cells LC2029 bacteria prevent contact of C. albicans with epithelial cells and block the transition of the yeast non-pathogenic form to the pathogenic hyphal form. Autors revealed that Slp2 found on the surface of LC2029 bacteria provides LC2029 strain cells with the ability to co-aggregate with various strains of C. albicans, including clinical isolates. C. albicans-induced necrotizing epithelial damage is reduced by colonization with Slp2-positive 44 LC2029 strain. Correct methods were used in the research. The manuscript is written correctly. The discussed issue is of great practical and scientific importance. The research could be supported by more advanced electron microscopy techniques, e.g. SEM or AFM. Analysis of the cell surfaces would show an interesting effect. In my opinion, this documentation should be enriched. Both Candida albicans and HeLa cells are very good models for microscopic techniques.
Reviewer 3 Report
Comments and Suggestions for Authors
In this manuscript, the authors describe the activity of the Slp2-positive LC2029 strain and soluble Slp2 against C. albicans infections.
Introduction
The authors should enhance their focus on Candida-associated vaginal infections and the contemporary therapeutic interventions currently available.
Furthermore, the authors should provide a more detailed description of the studies on L. crispatus.
Line 54: "C. albicans is a dimorphic fungus," this sentence is incorrect, and the authors need to make a correction.
Line 66: “superficial infections”, the authors need to make a correction.
Results and Discussion
The authors evaluate the decrease in adherence across diverse cellular substrates using Lactobacillus crispatus, an organism inherent to the vaginal environment, and soluble Slp2. L. crispatus is also regarded as a significant microbial biomarker because of its purported beneficial effects on vaginal health (Argentini, Chiara, et al. "Evaluation of modulatory activities of Lactobacillus crispatus strains in the context of the vaginal microbiota." Microbiology Spectrum 10.2 (2022): e02733-21). The emphasis should be directed towards elucidating the active role exerted at the vaginal level, with a more comprehensive exploration of the associated mechanistic aspects. The authors demonstrate that Slp2 inhibits adhesion. It would be crucial to investigate the underlying reasons for this inhibition.
Round 2
Reviewer 3 Report
Comments and Suggestions for Authors
The authors have improved the manuscript and clarified my doubts.